# Reproducibility Report
# Rigging the Lottery: Making All Tickets Winners

**Varun Sundar**
University of Wisconsin Madison
vsundar4@wisc.edu

**Rajat Vadiraj Dwaraknath**
Stanford University
rajatvd@stanford.edu

## Reproducibility Summary

**Scope of Reproducibility**

For a fixed parameter count and compute budget, the proposed algorithm (*RigL*) claims to directly train sparse networks that match or exceed the performance of existing dense-to-sparse training techniques (such as pruning). *RigL* does so while requiring constant Floating Point Operations (FLOPs) throughout training. The technique obtains state-of-the-art performance on a variety of tasks, including image classification and character-level language-modelling.

**Methodology**

We implement *RigL* from scratch in Pytorch using boolean masks to simulate unstructured sparsity. We rely on the description provided in the original paper, and referred to the authors' code for only specific implementation detail such as handling overflow in ERK initialization. We evaluate sparse training using *RigL* for WideResNet-22-2 on CIFAR-10 and ResNet-50 on CIFAR-100, requiring 2 hours and 6 hours respectively per training run on a GTX 1080 GPU.

**Results**

We reproduce *RigL*'s performance on CIFAR-10 within 0.1% of the reported value. On both CIFAR-10/100, the central claim holds—given a fixed training budget, *RigL* surpasses existing dynamic-sparse training methods over a range of target sparsities. By training longer, the performance can match or exceed iterative pruning, while consuming constant FLOPs throughout training. We also show that there is little benefit in tuning *RigL*'s hyper-parameters for every sparsity, initialization pair—the reference choice of hyperparameters is often close to optimal performance.

Going beyond the original paper, we find that the optimal initialization scheme depends on the training constraint. While the Erdos-Renyi-Kernel distribution outperforms Random distribution for a fixed parameter count, for a fixed FLOP count, the latter performs better. Finally, redistributing layer-wise sparsity while training can bridge the performance gap between the two initialization schemes, but increases computational cost.

**What was easy**

The authors provide code for most of the experiments presented in the paper. The code was easy to run and allowed us to verify the correctness of our re-implementation. The paper also provided a thorough and clear description of the proposed algorithm without any obvious errors or confusing exposition.

**What was difficult**

Tuning hyperparameters involved multiple random seeds and took longer than anticipated. Verifying the correctness of a few baselines was tricky and required ensuring that the optimizer's gradient (or momentum) buffers were sparse (or dense) as specified by the algorithm. Compute limits restricted us from evaluating on larger datasets such as Imagenet.

**Communication with original authors**

We had responsive communication with the original authors, which helped clarify a few implementation and evaluation details, particularly regarding the FLOP counting procedure.

Preprint. Under review.

# 1 Introduction

Sparse neural networks are a promising alternative to conventional dense networks—having comparatively greater parameter efficiency and lesser floating-point operations (FLOPs) (Han et al. [2016], Ashby et al. [2017], Srinivas et al. [2017]). Unfortunately, present techniques to produce sparse networks of commensurate accuracy involve multiple cycles of training dense networks and subsequent pruning. Consequently, such techniques offer no advantage over training dense networks, either computationally or memory-wise.

In the paper Evci et al. [2020], the authors propose *RigL*, an algorithm for training sparse networks from scratch. The proposed method outperforms both prior art in training sparse networks, as well as existing dense-to-sparse training algorithms. By utilising dense gradients only during connectivity updates and avoiding any global sparsity redistribution, *RigL* can maintain a fixed computational cost and parameter count throughout training.

As a part of the ML Reproducibility Challenge, we replicate *RigL* from scratch and investigate if dynamic-sparse training confers significant practical benefits compared to existing sparsifying techniques.

# 2 Scope of reproducibility

In order to verify the central claims presented in the paper we focus on the following target questions:

- Does *RigL* outperform existing sparse-to-sparse training techniques—such as SET (Mocanu et al. [2018]) and SNFS (Dettmers and Zettlemoyer [2020])—and match the accuracy of dense-to-sparse training methods such as iterative pruning (Zhu and Gupta [2018])?
- *RigL* requires two additional hyperparameters to tune. We investigate the sensitivity of final performance to these hyperparameters across a variety of target sparsities (Section 5.3).
- How does the choice of sparsity initialization affect the final performance for a fixed parameter count and a fixed training budget (Section 6.1)?
- Does redistributing layer-wise sparsity during connection updates (Dettmers and Zettlemoyer [2020]) improve *RigL*'s performance? Can the final layer-wise distribution serve as a good sparsity initialization scheme (Section 6.2)?

# 3 Methodology

The authors provide publicly accessible code[1] written in Tensorflow (Abadi et al. [2016]). To gain a better understanding of various implementation aspects, we opt to replicate *RigL* in Pytorch (Paszke et al. [2019]). Our implementation extends the open-source code[2] of Dettmers and Zettlemoyer [2020] which uses a boolean mask to simulate unstructured sparsity. Our source code is publicly accessible on Github[3] with training plots available on WandB[4] (Biewald [2020]).

**Mask Initialization** For a network with $L$ layers and total parameters $N$, we associate each layer with a random boolean mask of sparsity $s_l$, $l \in [L]$. The overall sparsity of the network is given by $S = \frac{\sum_l s_l N_l}{N}$, where $N_l$ is the parameter count of layer $l$. Sparsities $s_l$ are determined by the one of the following mask initialization strategies:

- **Uniform:** Each layer has the same sparsity, i.e., $s_l = S \; \forall l$. Similar to the original authors, we keep the first layer dense in this initialization.
- **Erdos-Renyi (ER):** Following Mocanu et al. [2018], we set $s_l \propto \left(1 - \frac{C_{\text{in}} + C_{\text{out}}}{C_{\text{in}} \times C_{\text{out}}}\right)$, where $C_{\text{in}}, C_{\text{out}}$ are the in and out channels for a convolutional layer and input and output dimensions for a fully-connected layer.
- **Erdos-Renyi-Kernel (ERK):** Modifies the sparsity rule of convolutional layers in ER initialization to include kernel height and width, i.e., $s_l \propto \left(1 - \frac{C_{\text{in}} + C_{\text{out}} + w + h}{C_{\text{in}} \times C_{\text{out}} \times w \times h}\right)$, for a convolutional layer with $C_{\text{in}} \times C_{\text{out}} \times w \times h$ parameters.

We do not sparsify either bias or normalization layers, since these have a negligible effect on total parameter count.

---

[1]https://github.com/google-research/rigl

[2]https://github.com/TimDettmers/sparse_learning

[3]https://github.com/varun19299/rigl-reproducibility

[4]https://wandb.ai/ml-reprod-2020

Table 1: **Test accuracy of reference and our implementations on CIFAR-10,** tabulated for three different sparsities. Note that the runs listed here do not use a separate validation set while training.

| Method | Ours | | | Original | | |
|---|---|---|---|---|---|---|
| Dense | 94.6 | | | 94.1 | | |
| | $1-s=0.1$ | $1-s=0.2$ | $1-s=0.5$ | $1-s=0.1$ | $1-s=0.2$ | $1-s=0.5$ |
| Static (ERK) | 91.6 | 93.2 | 94.3 | 91.6 | 92.9 | 94.2 |
| Pruning | 93.2 | 93.6 | 94.3 | 93.3 | 93.5 | 94.1 |
| RigL (ERK) | 93.2 | 93.8 | 94.4 | 93.1 | 93.8 | 94.3 |

**Mask Updates**  Every $\Delta T$ training steps, certain connections are discarded, and an equal number are grown. Unlike SNFS (Dettmers and Zettlemoyer [2020]), there is no redistribution of layer-wise sparsity, resulting in constant FLOPs throughout training.

**Pruning Strategy**  Similar to SET and SNFS, *RigL* prunes $f$ fraction of smallest magnitude weights in each layer. As detailed below, the fraction $f$ is decayed across mask update steps, by cosine annealing:

$$f(t) = \frac{\alpha}{2}\left(1 + \cos\left(\frac{t\pi}{T_{\text{end}}}\right)\right) \tag{1}$$

where, $\alpha$ is the initial pruning rate and $T_{\text{end}}$ is the training step after which mask updates are ceased.

**Growth Strategy**  *RigL*'s novelty lies in how connections are grown: during every mask update, $k$ connections having the largest absolute gradients among current inactive weights (previously zero + pruned) are activated. Here, $k$ is chosen to be the number of connections dropped in the prune step. This requires access to dense gradients at each mask update step. Since gradients are not accumulated (unlike SNFS), *RigL* does not require access to dense gradients at *every* step. Following the paper, we initialize newly activated weights to zero.

## 4 Experimental Settings

### 4.1 Model descriptions

For experiments on CIFAR-10 (Alex Krizhevsky [2009]), we use a Wide Residual Network (Zagoruyko and Komodakis [2016]) with depth 22 and width multiplier 2, abbreviated as WRN-22-2. For experiments on CIFAR-100 (Alex Krizhevsky [2009]), we use a modified variant of ResNet-50 (He et al. [2016]), with the initial $7 \times 7$ convolution replaced by two $3 \times 3$ convolutions (architecture details provided in the supplementary material).

### 4.2 Datasets and Training descriptions

We conduct our experiments on the CIFAR-10 and CIFAR-100 image classification datasets. For CIFAR-10, we use a train/val/test split of 45k/5k/10k samples. In comparison, the authors use no dedicated validation set, with 50k samples and 10k samples comprising the train set and test set, respectively. This causes a slight performance discrepancy between our reproduction and the metrics reported by the authors (dense baseline has a test accuracy of 93.4% vs 94.1% reported). However, our replication matches the paper's performance when 50k samples are used for the train set (Table 4). We use a validation split of 10k samples for CIFAR-100 as well.

On both datasets, we train models for 250 epochs each, optimized by SGD with momentum. Our training pipeline uses standard data augmentation, which includes random flips and crops. When training on CIFAR-100, we additionally include a learning rate warmup for 2 epochs and label smoothening of 0.1 (Goyal et al. [2017]). We also initialize the last batch normalization layer (Ioffe and Szegedy [2015]) in each BottleNeck block to 0, following He et al. [2019].

### 4.3 Hyperparameters

*RigL* includes two additional hyperparameters $(\alpha, \Delta T)$ in comparison to regular dense network training. In Sections 5.1 and 5.2, we set $\alpha = 0.3, \Delta T = 100$, based on the original paper. Optimizer specific hyperparameters—learning rate, learning rate schedule, and momentum—are also set according to the original paper. In Section 5.3, we tune these

Table 2: **WideResNet-22-2 on CIFAR10**, tabulated for two density $(1 - s)$ values. We group methods by their FLOP requirement and in each group, we mark the best accuracy in bold. Similar to Evci et al. [2020], we assume that algorithms utilize sparsity during training. All results are obtained by methods implemented in our unified codebase.

| Method | $1 - \mathbf{s} = \mathbf{0.1}$ | | $1 - \mathbf{s} = \mathbf{0.2}$ | |
| --- | --- | --- | --- | --- |
| | Accuracy ↑ (Test) | FLOPs ↓ (Train, Test) | Accuracy ↑ (Test) | FLOPs ↓ (Train, Test) |
| Small Dense | $89.0 \pm 0.35$ | 0.11x, 0.11x | $91.0 \pm 0.07$ | 0.20x, 0.20x |
| Static | $89.1 \pm 0.17$ | 0.10x, 0.10x | $91.2 \pm 0.16$ | 0.20x,0.20x |
| SET | $91.3 \pm 0.47$ | 0.10x, 0.10x | $\mathbf{92.7 \pm 0.28}$ | 0.20x, 0.20x |
| **RigL** | $\mathbf{91.7 \pm 0.18}$ | 0.10x, 0.10x | $92.6 \pm 0.10$ | 0.20x, 0.20x |
| SET (ERK) | $92.2 \pm 0.04$ | 0.17x, 0.17x | $92.9 \pm 0.16$ | 0.35x, 0.35x |
| **RigL (ERK)** | $\mathbf{92.4 \pm 0.06}$ | 0.17x, 0.17x | $\mathbf{93.1 \pm 0.09}$ | 0.35x, 0.35x |
| Static$_{2\times}$ | $89.15 \pm 0.17$ | 0.20x, 0.10x | $91.2 \pm 0.16$ | 0.40x, 0.20x |
| Lottery | $90.4 \pm 0.09$ | 0.45x, 0.13x | $92.0 \pm 0.31$ | 0.68x,0.27x |
| SET$_{2\times}$ | $83.3 \pm 15.33$ | 0.20x, 0.10x | $93.0 \pm 0.22$ | 0.41x, 0.20x |
| SNFS | $92.4 \pm 0.43$ | 0.51x, 0.27x | $92.7 \pm 0.20$ | 0.66x, 0.49x |
| SNFS (ERK) | $92.2 \pm 0.2$ | 0.52x, 0.28x | $92.8 \pm 0.07$ | 0.66x, 0.49x |
| SNFS$_{2\times}$ | $92.3 \pm 0.33$ | 1.02x, 0.27x | $93.2 \pm 0.14$ | 1.32x, 0.98x |
| RigL$_{2\times}$ | $92.3 \pm 0.25$ | 0.20x, 0.10x | $93.0 \pm 0.21$ | 0.41x, 0.20x |
| Pruning | $92.6 \pm 0.08$ | 0.32x,0.13x | $93.2 \pm 0.27$ | 0.41x,0.27x |
| **RigL$_{2\times}$ (ERK)** | $\mathbf{92.7 \pm 0.37}$ | 0.34x, 0.17x | $\mathbf{93.3 \pm 0.09}$ | 0.70x, 0.35x |
| **Dense Baseline** | $\mathbf{93.4 \pm 0.07}$ | 9.45e8, 3.15e8 | - | - |

hyperparameters with Optuna (Akiba et al. [2019]). We also examine whether indivdually tuning the learning rate for each sparsity value offers any significant benefit.

### 4.4 Baseline implementations

We compare *RigL* against various baselines in our experiments: SET (Mocanu et al. [2018]), SNFS (Dettmers and Zettlemoyer [2020]), and Magnitude-based Iterative-pruning (Zhu and Gupta [2018]). We also compare against two weaker baselines, viz., *Static Sparse* training and *Small-Dense* networks. The latter has the same structure as the dense model but uses fewer channels in convolutional layers to lower parameter count. We implement iterative pruning with the pruning interval kept same as the masking interval for a fair comparison.

### 4.5 Computational requirements

We run our experiments on a SLURM cluster node—equipped with 4 NVIDIA GTX1080 GPUs and a 32 core Intel CPU. Each experiment on CIFAR-10 and CIFAR-100 consumes about 1.6 GB and 7 GB of VRAM respectively and is run for 3 random seeds to capture performance variance. We require about 6 and 8 days of total compute time to produce all results, including hyper-parameter sweeps and extended experiments, on CIFAR-10 and CIFAR-100 respectively.

## 5 Results

Given a fixed training FLOP budget, *RigL* surpasses existing dynamic sparse training methods over a range of target sparsities, on both CIFAR-10 and 100 (Sections 5.1, 5.2). By training longer, *RigL* matches or marginally outperforms iterative pruning. However, unlike pruning, its FLOP consumption is constant throughout. This a prime reason for using sparse networks, and makes training larger networks feasible. Finally, as evaluated on CIFAR-10, the original authors' choice of hyper-parameters are close to optimal for multiple target sparsities and initialization schemes (Section 5.3).

### 5.1 WideResNet-22 on CIFAR-10

Results on the CIFAR-10 dataset are provided in Table 2. Tabulated metrics are averaged across 3 random seeds and reported with their standard deviation. All sparse networks use random initialization, unless indicated otherwise.

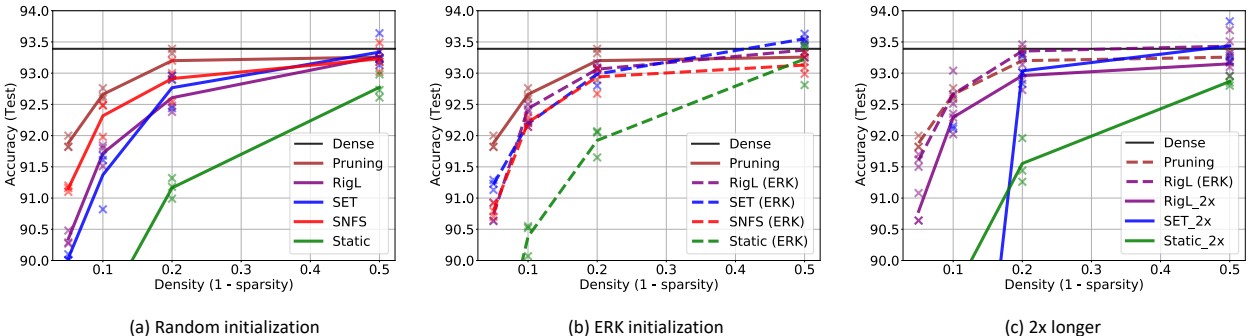

(a) Random initialization        (b) ERK initialization        (c) 2x longer

Figure 1: **Test Accuracy vs Sparsity on CIFAR-10,** plotted for Random initialization (**left**), ERK initialization (**center**), and for training $2\times$ longer (**right**). Owing to random growth, SET can be unstable when training for longer durations with higher sparsities. Overall, $RigL_{2\times}$ (ERK) achieves highest test accuracy.

While SET improves over the performance of static sparse networks and small-dense networks, methods utilizing gradient information (SNFS, *RigL*) obtain better test accuracies. SNFS can outperform *RigL*, but requires a much larger training budget, since it (a) requires dense gradients at each training step, (b) redistributes layer-wise sparsity during mask updates. For all sparse methods, excluding SNFS, using ERK initialization improves performance, but with increased FLOP consumption. We calculate theoretical FLOP requirements in a manner similar to Evci et al. [2020] (exact details in the supplementary material).

Figure 1 contains test accuracies of select methods across two additional sparsity values: $(0.5, 0.95)$. At lower sparsities (higher densities), *RigL* matches the performance of the dense baseline. Performance further improves by training for longer durations. Particularly, training *RigL* (ERK) twice as long at 90% sparsity exceeds the performance of iterative pruning while requiring similar theoretical FLOPs. This validates the original authors' claim that *RigL* (a sparse-to-sparse training method) outperforms pruning (a dense-to-sparse training method).

### 5.2 ResNet-50 on CIFAR100

Table 3 & Figure 2: **Benchmarking sparse ResNet-50s on CIFAR-100,** tabulated by performance and cost (**below**), and plotted across densities (**right**). In each group below, *RigL* outperforms or matches existing sparse-to-sparse and dense-to-sparse methods. Notably, $RigL_{3\times}$ at 90% sparsity and $RigL_{2\times}$ at 80% sparsity surpass iterative pruning with similar FLOP consumption. $RigL_{2\times}$ (ERK) further improves performance but requires a larger training budget.

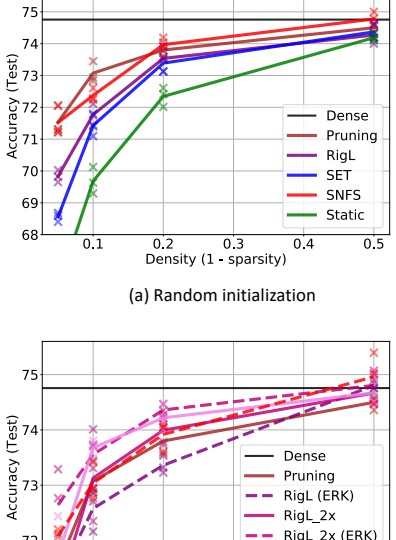

(a) Random initialization

(b) ERK initialization, Extended training

| Method | $1-s = 0.1$ | | $1-s = 0.2$ | |
|---|---|---|---|---|
| | Accuracy ↑ (Test) | FLOPs ↓ (Train, Test) | Accuracy ↑ (Test) | FLOPs ↓ (Train, Test) |
| Static | $69.7 \pm 0.42$ | 0.10x, 0.10x | $72.3 \pm 0.30$ | 0.20x,0.20x |
| Small Dense | $70.8 \pm 0.22$ | 0.11x, 0.11x | $72.6 \pm 0.93$ | 0.20x, 0.20x |
| SET | $71.4 \pm 0.35$ | 0.10x, 0.10x | $73.4 \pm 0.45$ | 0.20x, 0.20x |
| **RigL** | $\mathbf{71.8 \pm 0.33}$ | 0.10x, 0.10x | $\mathbf{73.5 \pm 0.04}$ | 0.20x, 0.20x |
| Static (ERK) | $71.5 \pm 0.18$ | 0.22x, 0.22x | $73.2 \pm 0.39$ | 0.38x, 0.38x |
| SET (ERK) | $72.3 \pm 0.39$ | 0.22x, 0.22x | $\mathbf{73.5 \pm 0.25}$ | 0.38x, 0.38x |
| **RigL (ERK)** | $\mathbf{72.6 \pm 0.37}$ | 0.23x, 0.22x | $73.4 \pm 0.15$ | 0.38x, 0.38x |
| SNFS | $72.3 \pm 0.20$ | 0.58x, 0.37x | $73.9 \pm 0.20$ | 0.70x, 0.55x |
| SNFS (ERK) | $73.0 \pm 0.33$ | 0.59x, 0.38x | $73.9 \pm 0.27$ | 0.69x, 0.54x |
| Pruning | $73.1 \pm 0.32$ | 0.36x,0.11x | $73.8 \pm 0.23$ | 0.45x,0.25x |
| $RigL_{2\times}$ | $73.1 \pm 0.71$ | 0.20x, 0.10x | $74.0 \pm 0.24$ | 0.41x, 0.20x |
| Lottery | $73.6 \pm 0.32$ | 0.62x,0.11x | $74.2 \pm 0.41$ | 0.81x,0.25x |
| **$RigL_{3\times}$** | $\mathbf{73.7 \pm 0.16}$ | 0.30x, 0.10x | $74.2 \pm 0.23$ | 0.61x, 0.20x |
| **$RigL_{2\times}$ (ERK)** | $73.6 \pm 0.05$ | 0.46x, 0.22x | $\mathbf{74.4 \pm 0.10}$ | 0.76x, 0.38x |
| **Dense Baseline** | $\mathbf{74.7 \pm 0.38}$ | 7.77e9, 2.59e9 | **-** | - |

We see similar trends when training sparse variants of ResNet-50 on the CIFAR-100 dataset (Table 3, metrics reported as in Section 5.1). We also include a comparison against sparse networks trained with the Lottery Ticket Hypothesis

(Frankle and Carbin [2018]) in Table 3—we obtain tickets with a commensurate performance for sparsities lower than 80%. Finally, the choice of initialization scheme affects the performance and FLOP consumption by a greater extent than the method used itself, with the exception of SNFS (groups 1 and 2 in Table 3).

## 5.3 Hyperparameter Tuning

Table 4: **Reference vs Optimal** $(\alpha, \Delta T)$ **on CIFAR-10.** Optimal hyperparameters are obtained by tuning with a TPE sampler in Optuna. The difference between the reference and optimal performance is small, indicating that there is not a significant benefit in tuning $(\alpha, \Delta T)$ individually for each initialization and sparsity configuration.

| Initialization | Density $(1 - s)$ | Reference | | Optimal | |
|---|---|---|---|---|---|
| | | $(\alpha, \Delta T)$ | Accuracy ↑ (Test) | $(\alpha, \Delta T)$ | Accuracy ↑ (Test) |
| Random | 0.1 | 0.3, 100 | $91.7 \pm 0.18$ | 0.197, 50 | $\mathbf{91.8 \pm 0.17}$ |
| Random | 0.2 | 0.3, 100 | $92.6 \pm 0.10$ | 0.448, 150 | $\mathbf{92.8 \pm 0.16}$ |
| Random | 0.5 | 0.3, 100 | $\mathbf{93.3 \pm 0.07}$ | 0.459, 550 | $93.3 \pm 0.18$ |
| ERK | 0.1 | 0.3, 100 | $\mathbf{92.4 \pm 0.06}$ | 0.416, 200 | $\mathbf{92.4 \pm 0.23}$ |
| ERK | 0.2 | 0.3, 100 | $\mathbf{93.1 \pm 0.09}$ | 0.381, 950 | $\mathbf{93.1 \pm 0.21}$ |
| ERK | 0.5 | 0.3, 100 | $93.4 \pm 0.14$ | 0.287, 500 | $\mathbf{93.8 \pm 0.06}$ |

$(\alpha, \Delta T)$ **vs Sparsities**   To understand the impact of the two additional hyperparameters included in *RigL*, we use a Tree of Parzen Estimator (TPE sampler, Bergstra et al. [2011]) via Optuna to tune $(\alpha, \Delta T)$. We do this for sparsities $(1 - s) \in \{0.1, 0.2, 0.5\}$, and a fixed learning rate of 0.1. Additionally, we set the sampling domain for $\alpha$ and $\Delta T$ as $[0.1, 0.6]$ and $\{50, 100, 150, ..., 1000\}$ respectively. We use 15 trials for each sparsity value, with our objective function as the validation accuracy averaged across 3 random seeds.

Table 4 shows the test accuracies of tuned hyperparameters. While the reference hyperparameters (original authors, $\alpha = 0.3, \Delta T = 100$) differ from the obtained optimal hyperparameters, the difference in performance is marginal,

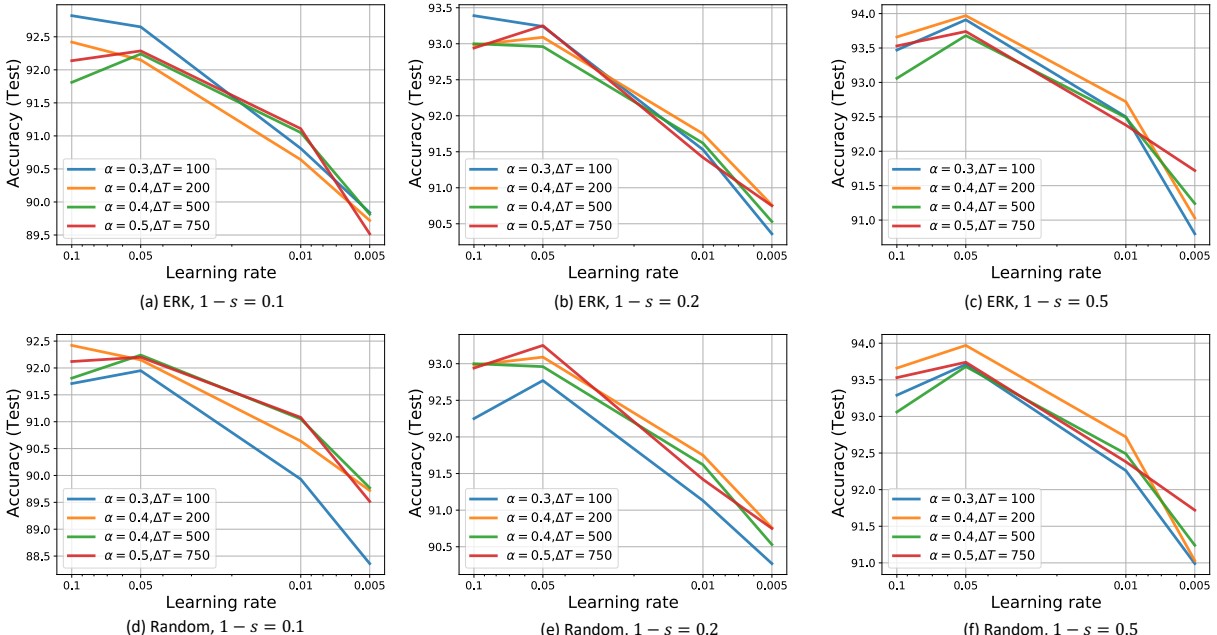

Figure 3: **Learning Rate vs Sparsity on CIFAR-10.** Runs using a learning rate $> 0.1$ do not converge and are not plotted here. There is little benefit in tuning the learning rate for each sparsity, and $0.1, 0.05$ are good choices overall.

especially for ERK initialization. This in agreement with the original paper, which finds $\alpha \in \{0.3, 0.5\}, \Delta T = 100$ to be suitable choices. We include contour plots detailing the hyperparameter trial space in the supplementary material.

**Learning Rate vs Sparsities** We further examine if the final performance improves by tuning the learning rate ($\eta$) individually for each sparsity-initialization pair. We employ a grid search over $\eta \in \{0.1, 0.05, 0.01, 0.005\}$ and $(\alpha, \Delta T) \in \{(0.3, 100), (0.4, 200), (0.4, 500), (0.5, 750)\}$. As seen in Figure 3, $\eta = 0.1$ and $\eta = 0.05$ are close to optimal values for a wide range of sparsities and initializations. Since these learning rates also correspond to good choices for the Dense baseline, one can employ similar values when training with *RigL*.

# 6 Results beyond Original Paper

## 6.1 Sparsity Distribution vs FLOP Consumption

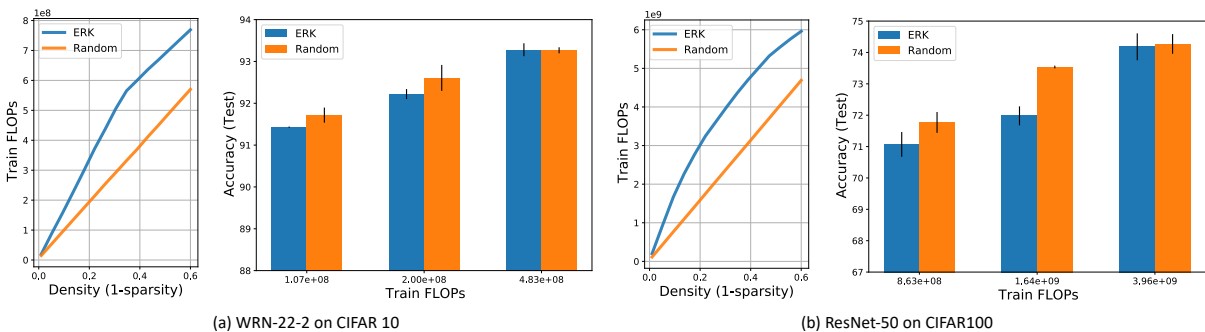

(a) WRN-22-2 on CIFAR 10      (b) ResNet-50 on CIFAR100

Figure 4: **Test Accuracy vs FLOP consumption of WideResNet-22-2 on CIFAR-10 and ResNet-50 on CIFAR-100,** compared for Random and ERK initializations. For the same FLOP budget, models trained with ERK initialization must be more sparse, resulting in inferior performance.

While ERK initialization outperforms Random initialization consistently for a given target parameter count, it requires a higher FLOP budget. Figure 4 compares the two initialization schemes across fixed training FLOPs. Theoretical FLOP requirement for Random initialization scales linearly with density $(1 - s)$, and is significantly lesser than ERK's FLOP requirements. Consequently, Random initialization outperforms ERK initialization for a given training budget.

## 6.2 Effect of Redistribution

Table 5: **Effect of redistribution during *RigL* updates, evaluated on CIFAR10 and CIFAR100**. By utilising sparse gradient or sparse momentum based redistribution, *RigL* (Random) matches *RigL* (ERK)'s performance. Among Random and ERK initialized experiments, we mark the best metrics under each sparsity and dataset in bold.

| Method | Redistribution | CIFAR-10 | | | | CIFAR-100 | | | |
|---|---|---|---|---|---|---|---|---|---|
| | | $1-s = 0.1$ | | $1-s = 0.2$ | | $1-s = 0.1$ | | $1-s = 0.2$ | |
| | | Accuracy ↑ (Test) | FLOPs ↓ (Train, Test) | Accuracy ↑ (Test) | FLOPs ↓ (Train, Test) | Accuracy ↑ (Test) | FLOPs ↓ (Train, Test) | Accuracy ↑ (Test) | FLOPs ↓ (Train, Test) |
| | | | | Random Initialization | | | | | |
| RigL | - | $91.7 \pm 0.18$ | 0.10x, 0.10x | $92.9 \pm 0.10$ | 0.20x, 0.20x | $71.8 \pm 0.33$ | 0.10x, 0.10x | $73.5 \pm 0.04$ | 0.20x, 0.20x |
| RigL-SG | Sparse Grad | $\mathbf{92.2 \pm 0.17}$ | 0.28x, 0.28x | $92.7 \pm 0.25$ | 0.49x, 0.49x | $72.3 \pm 0.12$ | 0.36x,0.35x | $\mathbf{73.7 \pm 0.15}$ | 0.53x, 0.53x |
| RigL-SM | Sparse Mmt | $\mathbf{92.2 \pm 0.20}$ | 0.28x, 0.28x | $\mathbf{92.9 \pm 0.21}$ | 0.50x, 0.49x | $\mathbf{72.6 \pm 0.27}$ | 0.36x,0.36x | $\mathbf{73.7 \pm 0.35}$ | 0.53x, 0.53x |
| | | | | ERK Initialization | | | | | |
| RigL | - | $\mathbf{92.4 \pm 0.06}$ | 0.17x, 0.17x | $\mathbf{93.1 \pm 0.09}$ | 0.35x, 0.35x | $72.6 \pm 0.37$ | 0.23x, 0.22x | $73.4 \pm 0.15$ | 0.38x, 0.38x |
| RigL-SG | Sparse Grad | $92.1 \pm 0.19$ | 0.28x, 0.28x | $92.7 \pm 0.19$ | 0.49x, 0.49x | $\mathbf{73.0 \pm 0.13}$ | 0.37x,0.36x | $74.2 \pm 0.26$ | 0.53x, 0.53x |
| RigL-SM | Sparse Mmt | $\mathbf{92.27 \pm 0.01}$ | 0.28x, 0.28x | $\mathbf{93.0 \pm 0.13}$ | 0.50x, 0.49x | $72.6 \pm 0.27$ | 0.37x, 0.37x | $\mathbf{74.2 \pm 0.13}$ | 0.53x, 0.53x |
| | | | | Re-Initialization with *RigL*-SM (Random, ERK) | | | | | |
| RigL | - | $90.3 \pm 0.34$ | 0.28x, 0.28x | $91.0 \pm 0.38$ | 0.50x, 0.49x | $67.6 \pm 0.28$ | 0.36x, 0.36x | $68.9 \pm 0.65$ | 0.53x, 0.53x |
| RigL (ERK) | - | $90.2 \pm 0.57$ | 0.28x, 0.28x | $90.6 \pm 0.56$ | 0.50x, 0.49x | $67.8 \pm 0.73$ | 0.37x, 0.37x | $68.9 \pm 0.47$ | 0.53x, 0.53x |

One of the main differences of *RigL* over SNFS is the lack of layer-wise redistribution during training. We examine if using a redistribution criterion can be beneficial and bridge the performance gap between Random and ERK initialization.

Following Dettmers and Zettlemoyer [2020], during every mask update, we reallocate layer-wise density proportional to its average sparse gradient or momentum (*RigL*-SG, *RigL*-SM).

Table 5 shows that redistribution significantly improves *RigL* (Random), but not *RigL* (ERK). We additionally plot the FLOP requirement against training steps and the final sparsity distribution in Figure 5. The layer-wise sparsity distribution largely becomes constant within a few epochs. The final distribution is similar, but more "extreme" than ERK—wherever ERK exceeds/falls short of Random, redistribution does so by a greater extent.

By allocating higher densities to $1 \times 1$ convolutions (*convShortcut* in Figure 5), redistribution significantly increases the FLOP requirement—and hence, is not a preferred alternative to ERK. Surprisingly, initializing *RigL* with the final sparsity distribution in a manner similar to the Lottery Ticket Hypothesis results in subpar performance (group 3, Table 5).

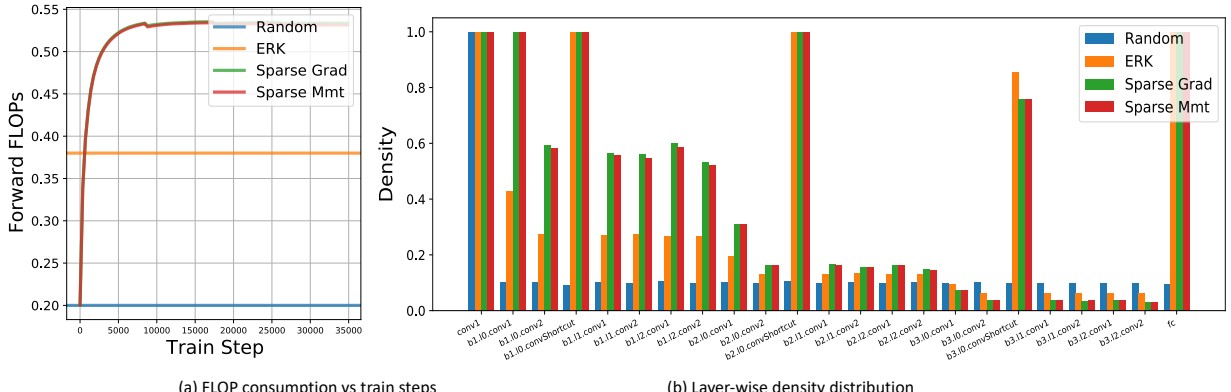

(a) FLOP consumption vs train steps          (b) Layer-wise density distribution

Figure 5: **Effect of redistribution on *RigL*'s performance,** evaluated using WideResNet-22-2 on CIFAR10 at 80% sparsity. **(left)** FLOPs required per forward pass, shown relative to the dense baseline, rises quickly and saturates within a few epochs (~10k steps) for both sparse gradient and sparse momentum based redistribution. **(right)** Comparison of the final density distribution against Random and ERK counterparts. "b" refers to block and "l" layer here.

# 7 Discussion

Evaluated on image classification, the central claims of Evci et al. [2020] hold true—*RigL* outperforms existing sparse-to-sparse training methods and can also surpass other dense-to-sparse training methods with extended training. *RigL* is fairly robust to its choice of hyperparameters, as they can be set independent of sparsity or initialization. We find that the choice of initialization has a greater impact on the final performance and compute requirement than the method itself. Considering the performance boost obtained by redistribution, proposing distributions that attain maximum performance given a FLOP budget could be an interesting future direction.

For computational reasons, our scope is restricted to small datasets such as CIFAR-10/100. *RigL*'s applicability outside image classification—in Computer Vision and beyond (machine translation etc.) is not covered here.

**What was easy**    The authors' code covered most of the experiments in their paper and helped us validate the correctness of our replicated codebase. Additionally, the original paper is quite complete, straightforward to follow, and lacked any major errors.

**What was difficult**    Implementation details such as whether momentum buffers were accumulated sparsely or densely had a substantial impact on the performance of SNFS. Finding the right $\epsilon$ for ERK initialization required handling of edge cases—when a layer's capacity is exceeded. Hyperparameter tuning $(\alpha, \Delta T)$ involved multiple seeds and was compute-intensive.

**Communication with original authors**    We acknowledge and thank the original authors for their responsive communication, which helped clarify a great deal of implementation and evaluation specifics. Particularly, FLOP counting for various methods while taking into account the changing sparsity distribution. We also discussed experiments extending the original paper—as to whether the authors had carried out a similar study before.

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
