# OpenReview forum: "[Reproducibility Report] Rigging the Lottery: Making All Tickets Winners"
_ML_Reproducibility_Challenge/2020 — RC2020_

### Official Review · AnonReviewer2 · 2021-02-18
**Good Reproducibility Project**

**Rating:** 7
**Confidence:** 3

**Review:**

The authors have chosen to reproduce results related to 'Rigging the Lottery: Making All Tickets Winners'. The code was based on that of the authors of the original paper. Results are consistent with the original ones for similar testbenches. However, tuning hyperprameters turned out to be challenging.

The reproducibility report is very well written, the problem is first formulated, the methodology is clearly presented, and the results are well described. The authors also indicate communications with those of the original paper, which is good to see.

**Familiar With The Original Paper:**

I have not read the original paper

**Reproducibility Summary:**

Report has summary

---

### Official Review · AnonReviewer4 · 2021-03-14
**Solid reproducibility report, strong accept!**

**Rating:** 10
**Confidence:** 4

**Review:**

Overall, this report is of very high quality and impact. The authors reproduce the original paper from scratch. The authors perform hyperparam sensitivity study. The authors also perform extensive ablation experiments and uncover an interesting finding on the dependency of initialization. This project checks all components needed for a good reproducibility report, and I believe this is worthy of a journal submission. I would like to thank the authors for their hard work!

* Reproducibility Summary

  The report contains a well-written, concrete reproducibility summary. The summary outlines the scope of the paper to reproduce the RigL algorithm from scratch by re-implementing the methodology on PyTorch. The summary also concisely highlights the major findings, where the report gets within 0.1% of the reported values in the original paper on CIFAR10.

* Scope of reproducibility

  The report investigates several central claims from the original paper, Riga. The report contains an investigation of the sensitivity of hyperparameters, model ablation, and choice of initialization too.

* Code: whether reproduced from scratch or re-used author repository.

  Authors re-implement the code of RigL from scratch in Pytorch, which was original written in Tensorflow. This makes the report extremely strong, as it helps to robustly validate the core claims of the original paper. The authors provide their code in the supplementary material. It is also very much appreciated that the authors plan to release the training plots, which would be a strong contribution towards the understanding of RigL.

* Communication with original authors

  Authors of the report communicated with the original authors successfully, who helped the authors clarify implementation and evaluation details.

* Hyperparameter Search

  Authors tune the hyperparameters with Optuna and carefully examine the impact of each hyperparam chosen in the original paper.

* Ablation Study

  The report goes a step further to perform an ablation study to investigate the impact of ERK initialization for a given target parameter count and training budget. The paper also performs experiments on redistribution, which is shown to help RigL with random initialization but not ERK. This is a very interesting finding!

* Discussion on results

  The report contains ample discussion on the results. Primarily, they find the central claim of the original paper holds true. RigL is also found to be fairly robust to the choice of hyperparameters. The report finds further evidence that the choice of initialization has a much greater impact on the final performance, and proposes interesting future directions for research.

* Recommendations for reproducibility

  The authors highly commend the original paper on their state of reproducibility and thank the original paper authors for their communication.

* Overall organization and clarity

  The paper is well organized and well written.

**Familiar With The Original Paper:**

I have not read the original paper

**Reproducibility Summary:**

Report has summary

---

### Decision · Program_Chairs · 2021-03-31

**Decision:**

Accept

**Comment:**

Selected for ReScience-C Journal Publication. This paper not only reproduced the results from the original paper, but added significant additional experiments by tuning hyperparameters, including evaluating different initialization schemes. They find that some distributions outperform others when the parameter count is fixed, but that conclusion is flipped when the FLOP count is fixed instead of the parameter count.